# Dry-Land Force–Velocity, Power–Velocity, and Swimming-Specific Force Relation to Single and Repeated Sprint Swimming Performance

**DOI:** 10.3390/jfmk8030120

**Published:** 2023-08-16

**Authors:** Ioannis Chalkiadakis, Gavriil G. Arsoniadis, Argyris G. Toubekis

**Affiliations:** 1Division of Aquatic Sports, School of Physical Education and Sports Science, National and Kapodistrian University of Athens, 17237 Athens, Greece; giannischalk@phed.uoa.gr (I.C.); garsoniadis@phed.uoa.gr (G.G.A.); 2Sports Performance Laboratory, School of Physical Education and Sports Science, National and Kapodistrian University of Athens, 17237 Athens, Greece

**Keywords:** strength, impulse, tethered swimming, performance, kinematical characteristics

## Abstract

The aim of this study was to identify the relationship between dry-land and in-water strength with performance and kinematic variables in short-distance, middle-distance, and repeated sprint swimming. Fifteen competitive swimmers applied a bench press exercise to measure maximum strength (MS), maximum power (P), strength corresponding to P (F@P), maximum velocity (MV), and velocity corresponding to P (V@P) using F–V and P–V relationships. On a following day, swimmers performed a 10 s tethered swimming sprint (TF), and impulse was measured (IMP). On three separate days, swimmers performed (i) 50 and 100 m, (ii) 200 and 400 m, and (iii) 4 × 50 m front crawl sprint tests. Performance time (T), arm stroke rate (SR), arm stroke length (SL), and arm stroke index (SI) were calculated in all tests. Performance in short- and middle-distance tests and in 4 × 50 m training sets were related to dry-land MS, P, TF, and IMP (r = 0.51–0.83; *p* < 0.05). MS, P, and TF were related to SR in 50 m and SI in 50 and 100 m (r = 0.55–0.71; *p* < 0.05). A combination of dry-land P and in-water TF variables explains 80% of the 50 m performance time variation. Bench press power and tethered swimming force correlate with performance in short- and middle-distance tests and repeated sprint swimming.

## 1. Introduction

Dry-land maximum strength and in-water-specific strength should be considered to improve swimmers’ performance [1]. Therefore, several dry-land [2,3] and in-water tests have been extensively used for strength evaluation [4,5,6]. A negative relationship has been observed between dry-land maximum strength and power with short-distance swimming performance time [3,7,8,9]. Several exercises, muscle groups, including upper and lower limbs, and contraction velocities have been used to examine this relationship and applied to various age groups of swimmers [7,8,9]. Emphasizing the dry-land evaluation, a force–velocity (F–V) and a power–velocity (P–V) profile may be used in testing possible relation to swimming performance [10,11]. An F–V and P–V profile, besides maximum strength, may provide additional variables such as speed of movement at various loads, maximum power and the corresponding load, or the power produced at a specific load. Thus, this helps in evaluating the functioning of muscles used during swimming and may be related to performance in a single or repeated sprint swimming effort.

Despite dry-land evaluation, a swimmer’s ability to produce high levels of in-water specific force during swimming is equally important. Tethered swimming tests, either of short (10 to 30 s) or long duration (120 s), can be applied with the aim of measuring in-water force or impulse and, consequently, its relationship with performance [4,5,6]. Regardless of tethered swimming test duration, a negative relation between in-water force and performance time at 50 and 100 m distances has been reported [2,8,12,13]. Moreover, the impulse is a variable that expresses the ability to produce in-water force to overcome hydrodynamic resistance, and it is related to short-distance [2,5] and middle-distance performance (200 m) [6,14]. Despite the strong correlation of specific dry-land strength or power and in-water force variables with swimming performance, there is no information concerning any connection of force or power requirements to high-intensity repeated sprint swimming training sets.

Swimming performance is also associated with technique [1], and it is important to test the possible relation between dry-land and in-water force with kinematic variables. Previous studies identified a connection between maximum strength and power and kinematic characteristics in short-distance tests (25 and 50 m front crawl) [3,9]. As far as in-water force, it seems to be correlated only with arm stroke index (SI) but not with arm stroke rate (SR) and arm stroke length (SL) in short-distance tests [12]. Nevertheless, there are no findings concerning dry-land variables obtained by F–V and P–V tests and in-water force correlations with kinematic characteristics during middle-distance tests or during a repeated sprint swimming training set. 

The purpose of the current study was to investigate the relationships between dry-land variables obtained by an F–V and P–V profile and in-water force variables with swimming performance and kinematics in short- (50, 100 m) and middle-distance tests (200, 400 m) and a repeated sprint swimming training set (4 × 50 m). We hypothesized that force in or out of water and land power characteristics would correlate with a swimmer’s performance time and kinematic variables during short-distance, middle-distance, and repeated sprint tests.

## 2. Materials and Methods

### 2.1. Participants 

Fifteen competitive swimmers (9 males and 6 females) volunteered to participate in the study. Swimmers’ anthropometrics and performance characteristics are shown in Table 1. Each swimmer was free from injury, and no medication was used prior to or during experimental procedures. All participants were required to have at least 5 years of experience in competitive swimming while participating in six swimming and one or two dry-land sessions per week as inclusion criteria. After a thorough explanation of the study procedures, all swimmers or their legal guardians signed a consent form that accepted their participation in the study. The local institutional review board approved the experimental protocol (approval number: 1111), which was according to the Helsinki Declaration.

### 2.2. Study Design 

Measurements were conducted in five testing sessions 24 h apart and lasted 1 week for each participant. Swimmers performed the following: (i) a test to measure maximum strength in bench press exercise, (ii) a 10 s tethered swimming sprint test to evaluate in-water force and impulse, (iii) 50 and 100 m front crawl sprints, (iv) 200 and 400 m maximum effort front crawl tests, and (v) four 50 m front crawl sprints (Figure 1). All tests were completed at the same time of the day (17:00 to 19:00 p.m.) and were applied in a 50 m outdoor swimming pool with a constant water temperature of 27 °C. Ambient temperature during testing ranged between 18 and 22 °C. Dry-land strength evaluation was conducted in an indoor gymnasium with an ambient temperature of 20–21 °C. The measurements were carried out by experienced and certified strength and conditioning researchers and professional swimming coaches. All swimmers were familiarized with bench press exercise at high loads and tethered swimming at maximum intensity in a separate session a week before the commencement of testing. A low-intensity aerobic swimming training and no dry-land exercise were applied the day before each testing session.

### 2.3. Dry-Land Strength and Tethered Force Evaluation 

#### 2.3.1. Maximum Strength Test in Bench Press 

Following a standardized warm-up (arm swings, medicine ball throws and a set of 12 to 15 repetitions with light load), each swimmer performed a bench press exercise on a Smith machine to measure one repetition maximum (1-RM) to be used as an index of maximum strength (MS). During the bench press exercise, the load was gradually increased in 3-to-5 set increments until 1-RM was achieved. A resting interval of about 4 min was applied after each set. A previously calibrated linear encoder connected with an analog-to-digital converter was attached to the barbell during each effort, and it was used to measure barbell displacement and speed of movement, which were used to draw the individual F–V and P–V relationships (MuscleLab, Ergotest, Stathelle, Norway). From F–V and P–V relationships, the maximum power (P), the force corresponding to maximum power (F@P), the maximum displacement velocity (MV), and the velocity displacement corresponding to maximum power (V@P) were calculated [15] (Figure 2).

#### 2.3.2. Tethered Swimming Test 

After 800 m standardized warm-up (400 m slow front crawl swimming, 4 × 50 m front crawl drills, and 4 × 50 m front crawl swim with progressively increasing speed), swimmers performed a 10 s front crawl tethered swimming sprint test to evaluate in-water force (TF). Force was measured with a sampling frequency of 100 Hz using a previously calibrated piezoelectric force transducer connected to an analog-to-digital converter (MuscleLab, Ergotest, Stathelle, Norway). The force transducer was attached to the hip of the swimmer using a non-elastic cable. A signal was inserted at the moment of each upper limb entry into the water to separate the arm strokes, allowing force and duration analysis for each arm stroke separately [16]. The impulse (IMP) of each arm stroke was calculated by the force and time using Equation (1):(1)$$\mathrm{IMP}=\int_{t2}^{t1}F\times dt,$$
where IMP is the impulse, and F is the mean applied force during each arm stroke or cycle from t1 to t2 (the time the swimmer’s hand is in the underwater phase) (Figure 3).

### 2.4. Swimming Performance Tests and Training Set 

During the third and fourth sessions, swimmers performed 50 and 100 m and 200 and 400 m front crawl swimming, respectively, applying maximum effort. Resting interval period between tests was 30 min. In the last session, swimmers completed a swimming training set, which consisted of four 50 m front crawl sprints (4 × 50 m) using a push-off start every 2 min. The mean performance time of 4 × 50 m was used for the statistical analysis. Before each swimming test, all participants performed the same warm-up as described previously for the tethered swimming test. Kinematic variables were measured in all swimming performance tests and during 4 × 50 m. Specifically, SR was measured each 50 m by the time taken to complete 3 consecutive stroke cycles, and SL was calculated by the ratio of swimming speed to SR each 50 m. The product of SL and swimming speed was used for calculation of SI during all tested distances and 4 × 50 m training sets. 

### 2.5. Statistical Analysis 

Normal distribution of the data was tested using the Kolmogorov–Smirnov test. Pearson correlation was used to examine relationships between swimming performance, kinematic, and strength variables in and out of water. Multiple linear regression analysis using a stepwise method was applied to identify combination of significant variables that may explain swimming performance during 50, 100, 200, and 400 m or during 4 × 50 m sprints. A priori analysis for sample size and statistical power using a correlation bivariate model was used [17]. Considering a critical r = 0.52 and a statistical power of 0.66, a sample size of fifteen participants (N= 15) was required. The 95% confidence limits were also calculated for each variable. Data are presented as mean ± SD. Statistical significance was set at *p* < 0.05.

## 3. Results

### 3.1. Swimming Performance 

#### 3.1.1. Dry-Land Strength Variables and Swimming Performance 

Descriptive statistics of all variables are presented in Table 2. Dry-land strength variables that were calculated during the bench press exercise (MS, P, F@P) were negatively correlated (r = −0.52 to −0.78, *p* < 0.05) with swimmers’ performance time during short (50, 100 m) and middle distances (200, 400 m) and the 4 × 50 m training sets (Table 3; note that negative correlations appear since a better performance indicates shorter time to complete a fixed distance). Likewise, V@P was negatively correlated (r = −0.58 to −0.80, *p* < 0.05) with performance time in all swimming distances and training sets, while MV was negatively correlated only with performance time in the 50, 100, and 200 m swimming (r = −0.55 to −0.65, *p* < 0.05). The stepwise multiple regression analysis showed that a combination of dry-land (P) and in-water (TF) variables may explain 80% of the variance in the 50 m performance time (R^2^ = 0.80, *p* < 0.01).

#### 3.1.2. In-Water Strength Variables and Swimming Performance 

In-water strength variable TF was negatively correlated with performance time in the 50 and 100 m swimming (r = −0.66 to −0.84, *p* < 0.05), as well as in the 200, 400, and 4 × 50 m swimming (r = −0.66 to −0.76, *p* < 0.05, Table 3). Moreover, the 50 m performance time was 76% predicted by TF (*p* = 0.01). 

#### 3.1.3. Impulse and Swimming Performance 

During the tethered swimming test, IMP was correlated with swimming performance time in the 50, 100, 200, and 4 × 50 m performance times (r = −0.57 to −0.71, *p* < 0.05, Table 3). 

### 3.2. Swimming Technique 

#### 3.2.1. Dry-Land Strength and Kinematic Variables 

There was positive relation between P and SR_50_ (r = 0.55; *p* < 0.05, Table 4). On the contrary, SI_50_ and SI_100_ presented a positive correlation with MS, P, and F@P (r = 0.56 to 0.68, *p* < 0.05, Table 4). MS, P, and F@P, however, did not correlate with SL in any swimming distance (*p* > 0.05, Table 4). Likewise, V@P and MV correlated positively with SI_50_, SI_100_, and SI_200_ (r = 0.64 to 0.69, *p* < 0.05, Table 4).

#### 3.2.2. In-Water Strength and Kinematic Variables 

It was found that TF and IMP had a positive relation with SR_50_ (r = 0.66 to 0.69, *p* < 0.05, Table 4). Furthermore, IMP was related to mean SR_4 × 50_ (r = 0.53, *p* < 0.05, Table 4). Moreover, all the in-water strength variables were correlated with SI_50_ (see Table 4). However, only TF was related to SI_100_ and SI_400_ (Table 4). 

## 4. Discussion

The purpose of the current study was to investigate the relationship of dry-land variables obtained by F–V and P–V relationships and in-water force with swimming performance time at various distances and during a repeated sprint swimming training set. Strong relationships both for the dry-land bench press exercise and in-water force variables with performance time in short- (50, 100 m) and middle-distance tests (200, 400 m) and 4 × 50 m training sets were observed. Furthermore, a positive relationship of SI in 50, 100, and 200 m distances with dry-land P and V@P was observed. In addition, in-water TF was correlated with SR in the 50 m and SI in the 50, 100, and 400 m front crawl performance tests. 

Dry-land bench press strength variables obtained by an F–V curve such as MS, P, F@P, and V@P were correlated with performance time in all front crawl tests and with a 4 × 50 m repeated sprint test time. These findings seem to agree with those reported in previous research studies using short-distance swimming events and dry-land strength variables [18]. Specifically, a higher negative correlation, compared to the current findings, has been demonstrated between performance time in the 50 m front crawl with maximum power (r = −0.76) and maximum strength (r = −0.77) despite the different dry-land exercise (pull-ups) that swimmers applied [18]. A higher correlation of −0.96 was reported when MS was evaluated using a push-up exercise [3], while a lower correlation with the 50 m performance time was observed using latissimus pull-down back average power in a 30 s test [8]. Such inconsistencies highlight the importance of applying sport-specific dry-land exercises for swimmers’ training [19]. Likewise, V@P and MV correlated negatively with swimming performance time in the short-distance tests (see Table 3). The speed of movement in the throwing tests correlates with 50 m sprint performance [7], and it may partially explain how performance time depends on swimmers’ muscle capacity (i.e., muscle fiber recruitment and force maintenance during a test [2,20]). 

Considering middle-distance performance tests (200, 400 m), a relationship between dry-land maximum power and performance time has recently been verified [21]. Nonetheless, it should be mentioned that the dry-land exercises that swimmers applied in previous studies (medicine ball throwing and horizontal jump) differed from those reported in the current study. It seems that MS, F@P, and P are contributing factors not only in short-distance but also in middle-distance events. Moreover, the high correlation with performance in a 4 × 50 m sprint training set indicates its supporting value in high-intensity training. Eventually, it seems that V@P and MV are not related to SR in any swimming test, but V@P was correlated with SI in the 50, 100, and 200 m tests, indicating that a specific fraction of maximum speed of contraction is required for effective muscle function in swimming. Thus, the MV obtained by the F–V test may not correlate with swimming upper limbs’ speed of movement (i.e., SR). The exercise selected in the present study (bench press) activates a similar muscle group to that used in the front crawl [22], but may not facilitate muscle actions at a relevant speed. Recent evidence suggests that the speed of movement, as well as the performance level of swimmers, may alter the correlation with swimming speed [23]. This limitation may be partially overcome using the F–V and P–V profiles in specific dry-land exercises. This is supported by the high correlation of V@P and F@P obtained in the present study. Nonetheless, the present findings highlight the necessity to schedule dry-land strength training sessions to increase swimmers’ strength and power. 

Dry-land strength, TF, and IMP correlated negatively with performance time in 50 m swimming, as it has previously mentioned [5,24]. In addition, Morouço et al. [25] found similar correlations between TF and swimming velocity in short-distance tests, although the tethered swimming test was of longer duration (30 s). It is possible that the ability to develop muscular force rapidly and the force by time product in the water influences performance in a short-distance test [2]. In addition, the specificity of in-water force tests with swimming movement patterns may explain this association. Considering middle-distance performance tests (200, 400 m), a relationship between in-water TF and performance time has recently been reported [13,14,25]. However, the tethered swimming time of 30 to 120 s was longer compared to the present study. The reported relationships in short-distance tests with TF and IMP were greater compared to the corresponding relationships with 200 and 400 m tests, but comparable to those observed in a 4 × 50 m training set. Such a variation may be attributed to different energetic demands and arm stroke characteristics during middle distances [2]. Whatever the case, the present study indicates that not only a single maximum effort but also repeated sprints, such as those used in high-intensity training, relate to dry-land and in-water-specific strength.

Considering the kinematic variables, SI in short-distance tests was related to dry-land- and in-water-evaluated variables. Similarly, a positive relationship between P in the bench press exercise and swimmers’ SR in the 50 m and 4 × 50 m repeated sprint tests was observed. Likewise, it should be mentioned that TF was positively related to SI in 400 m. SI is a kinematic variable that is associated with swimming efficiency, especially for middle distances [1]. Despite the limited information in the literature, it is likely that swimmers may benefit by improving in-water-specific strength, including in-water resisted training [26].

Finally, P and TF may explain 80% of the swimmer’s 50 m performance time, and TF may explain 76% of the mean performance time during a 4 × 50 m training test. Moreover, a positive relation of SR during the 50 m test with all in-water strength variables was observed (see Table 4). Based on this information, coaches need to construct a training plan to increase dry-land and in-water strength, especially in swimmers who compete in short performance distances (50 and 100 m). There are some limitations of the present study that need to be mentioned. Both male and female swimmers participated in the current study, and a longer duration tethered swimming test may possibly be used to increase relations with middle-distance tests. Moreover, the use of additional dry-land exercises using other muscle groups (i.e., latissimus dorsi) may reveal connections and relationships not shown in the present data. Future studies may focus on the relationship of strength and power in different periods of swimming training preparation.

## 5. Conclusions

The main findings of the present study indicate a significant relationship between dry-land strength and maximum power in the bench press and performance in short- and middle-distance tests. Likewise, the estimated variables from the F–V and P–V curves indicated a significant relation with performance time in short-distance tests. Furthermore, strength and impulse during 10 s tethered swimming are related to performance in short and middle-performance tests. Also, dry-land and in-water variables were related to kinematic parameters in short-distance tests. Our findings suggest that strength and power produced by swimmers in exercises out of the water, as well as the maximum velocity and the velocity at maximum power obtained by F–V and P–V curves or in tethered swimming, are connected with swimming performance in short- and middle-distance tests. 

## Figures and Tables

**Figure 1 jfmk-08-00120-f001:**
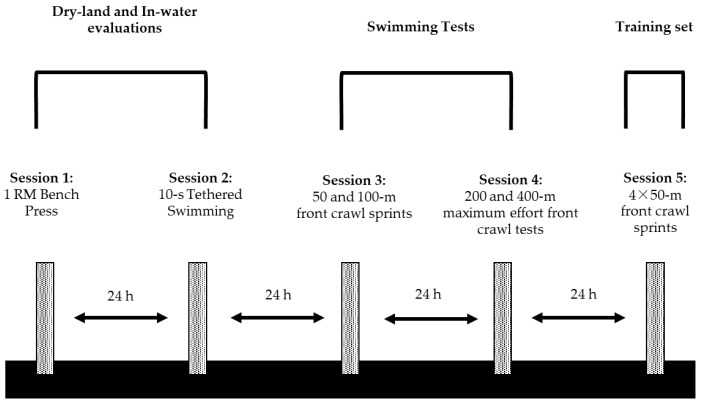
Experimental design of the study; 1-RM: one repetition maximum and h: hours.

**Figure 2 jfmk-08-00120-f002:**
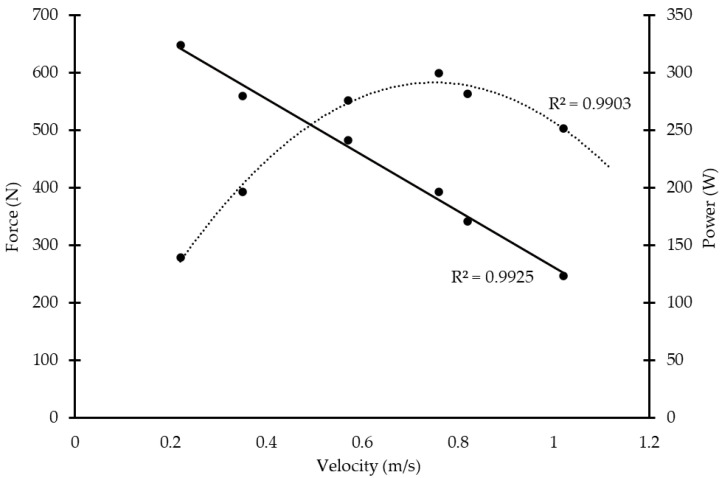
Force–velocity and power–velocity relationship to predict maximum power in bench press exercise. Example of one participant.

**Figure 3 jfmk-08-00120-f003:**
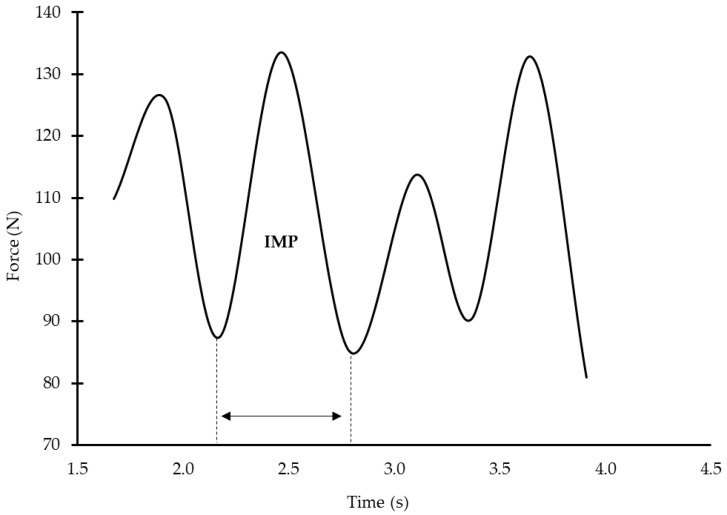
Example of force output and impulse calculation during consecutive arm strokes during tethered swimming. IMP: impulse.

**Table 1 jfmk-08-00120-t001:** Anthropometric and performance characteristics of the participants.

Variables	Overall (n = 15)	Male (n = 9)	Female (n = 6)
Age (yrs.)	16.7 ± 3.1	17.3 ± 3.6	15.7 ± 1.9
Body mass (kg)	60.7 ± 8.3	62.4 ± 9.8	58.2 ± 5.0
Body height (cm)	170.3 ± 9.3	173.3 ± 9.7	165.8 ± 7.4
Body mass index (kg/m^2^)	20.9 ± 1.8	20.6 ± 2.1	21.2 ± 1.3
Body fat (%)	16.3 ± 4.4	14.1 ± 3.9	19.7 ± 2.8
100 m front crawl performance time (s)	71.37 ± 5.59	68.45 ± 4.30	75.76 ± 4.40
FINA points (100 m front crawl)	386.8 ± 118.3	392.0 ± 114.5	378.4 ± 137.3
Competitive training experience (yrs.)	7.9 ± 1.6	8.2 ± 1.9	7.5 ± 1.1

FINA: Fédération Internationale de Natation Amateur.

**Table 2 jfmk-08-00120-t002:** Descriptive statistics of swimmers’ dry-land strength, in-water strength variables, performance time, and kinematic characteristics. Data reported as mean values and standard deviations (Mean ± SD). The 95% confidence limit (CL) for each variable is reported in the table.

Variables	Mean ± SD (95% CL)
**Dry-Land variables**	
P (W)	132.17 ± 69.73 (96.88–167.46)
F@P (N)	353.95 ± 105.75 (300.48–407.51)
MS (N)	704.41 ± 211.20 (597.54–811.29)
V@P (m·s^−1^)	0.36 ± 0.11 (0.30–0.41)
MV (m·s^−1^)	0.74 ± 0.21 (0.63–0.85)
**In-water strength variables**	
TF (N)	89.93 ± 26.14 (76.70–103.15)
IMP (N·s^−1^)	28.96 ± 11.07 (23.36–34.56)
**Swimming Performance Variables**	
T_50_ (s)	31.84 ± 2.99 (30.33–33.35)
T_100_ (s)	71.37 ± 5.59 (68.54–74.20)
T_200_ (s)	149.88 ± 13.59 (143.00–156.75)
T_400_ (s)	318.47 ± 27.35 (304.63–332.32)
T_4×50_ (s)	33.81 ± 2.59 (32.50–35.12)
**Kinematic Variables**	
**50 m**	
SR_50_ (cycles·min^−1^)	46.59 ± 4.22 (44.46–48.73)
SL_50_ (m·cycle^−1^)	2.04 ± 0.14 (1.97–2.11)
SI_50_ (m^2^·s^−1^·cycle^−1^)	3.24 ± 0.43 (3.02–3.46)
**100 m**	
SR_100_ (cycles·min^−1^)	39.74 ± 4.35 (37.54–41.94)
SL_100_ (m·cycle^−1^)	2.15 ± 0.23 (2.03–2.26)
SI_100_ (m^2^·s^−1^·cycle^−1^)	3.03 ± 0.45 (2.80–3.26)
**200 m**	
SR_200_ (cycles·min^−1^)	37.68 ± 4.49 (35.41–39.96)
SL_200_ (m·cycle^−1^)	2.17 ± 0.23 (2.05–2.29)
SI_200_ (m^2^·s^−1^·cycle^−1^)	2.94 ± 0.41 (2.74–3.15)
**400 m**	
SR_400_ (cycles·min^−1^)	35.22 ± 3.37 (33.51–36.92)
SL_400_ (m·cycle^−1^)	2.18 ± 0.23 (2.06–2.30)
SI_400_ (m^2^·s^−1^·cycle^−1^)	2.79 ± 0.44 (2.57–3.01)
**4 × 50 m training sets**	
SR_4×50_ (cycles·min^−1^)	42.71 ± 4.48 (40.44–44.98)
SL_4×50_ (m·cycle^−1^)	2.11 ± 0.20 (2.00–2.21)
SI_4×50_ (m^2^·s^−1^·cycle^−1^)	3.14 ± 0.42 (2.93–3.35)

P: maximum power, F@P: force at maximum power, MS: maximum strength, V@P: velocity in maximum power, MV: maximum velocity, TF: tethered force, IMP: impulse, T: performance time, SR: arm stroke rate, SL: arm stroke length, and SI: arm stroke index.

**Table 3 jfmk-08-00120-t003:** Correlations for swimmer’s performance time in 50, 100, 200, 400, and 4 × 50 m front crawl with dry-land strength and power and in-water strength variables.

Variables	P	F@P	MS	V@P	MV	TF	IMP
T_50_	−0.71 *	−0.66 *	−0.59 *	−0.67 *	−0.55 *	−0.84 *	−0.71 *
T_100_	−0.77 *	−0.68 *	−0.62 *	−0.77 *	−0.61 *	−0.76 *	−0.64 *
T_200_	−0.78 *	−0.68 *	−0.61 *	−0.80 *	−0.65 *	−0.68 *	−0.57 *
T_400_	−0.64 *	−0.64 *	−0.56 *	−0.64 *	−0.45	−0.66 *	−0.48
T_4×50_	−0.61 *	−0.58 *	−0.52 *	−0.58 *	−0.40	−0.76 *	−0.65 *

P: maximum power, F@P: force at maximum power, MS: maximum strength, V@P: velocity in maximum power, MV: maximum velocity, TF: tethered force, IMP: impulse, T: performance time, * *p* < 0.05.

**Table 4 jfmk-08-00120-t004:** Correlations between tethered swimming and dry-land variables and 50, 100, 200, 400 m, and 4 × 50 m training set kinematic variables.

Variables	P	F@P	MS	V@P	MV	TF	IMP
SR_50_ (cycles·min^−1^)	0.55 *	0.42	0.35	0.45	0.47	0.69 *	0.66 *
SL_50_ (m·cycle^−1^)	0.26	0.34	0.34	0.34	0.17	0.23	0.11
SI_50_ (m^2^·s^−1^·cycle^−1^)	0.63 *	0.62 *	0.56 *	0.64 *	0.49	0.71 *	0.57 *
SR_100_ (cycles·min^−1^)	0.17	0.04	−0.02	0.16	0.25	0.27	0.32
SL_100_ (m·cycle^−1^)	0.37	0.42	0.45	0.37	0.18	0.25	0.11
SI_100_ (m^2^·s^−1^·cycle^−1^)	0.68 *	0.65 *	0.63 *	0.69 *	0.48	0.58 *	0.41
SR_200_ (cycles·min^−1^)	0.43	0.39	0.32	0.40	0.47	0.42	0.46
SL_200_ (m·cycle^−1^)	0.14	0.08	0.10	0.29	0.02	0.05	−0.07
SI_200_ (m^2^·s^−1^·cycle^−1^)	0.64 *	0.51	0.48	0.68 *	0.47	0.48	0.31
SR_400_ (cycles·min^−1^)	0.31	0.29	0.25	0.25	0.35	0.17	0.18
SL_400_ (m·cycle^−1^)	0.20	0.21	0.19	0.23	0.01	0.37	0.19
SI_400_ (m^2^·s^−1^·cycle^−1^)	0.50	0.51	0.45	0.48	0.27	0.63 *	0.38
SR_4×50_ (cycles·min^−1^)	0.31	0.24	0.18	0.21	0.34	0.42	0.53 *
SL_4×50_ (m·cycle^−1^)	0.12	0.15	0.17	0.12	0.09	0.12	−0.07
SI_4×50_ (m^2^·s^−1^·cycle^−1^)	0.42	0.41	0.38	0.40	0.17	0.50	0.30

P: maximum power, F@P: force at maximum power, MS: maximum strength, V@P: velocity in maximum power, MV: maximum velocity, TF: tethered force, IMP: impulse, T: performance time, SR: arm stroke rate, SL: arm stroke length, and SI: arm stroke index. * *p* < 0.05.

## Data Availability

The data will be made available upon reasonable request to the corresponding author.

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
