# Peer review of "Dry-Land Force–Velocity, Power–Velocity, and Swimming-Specific Force Relation to Single and Repeated Sprint Swimming Performance"

_jfmk, 2023, doi:10.3390/jfmk8030120_

Round 1
Reviewer 1 Report

The English must be improved
Author Response
Response to Reviewer 1
JFMK
Revised Manuscript ID: JFMK- 2511252. Paper title: Dry-land force–velocity, power-velocity and swimming specific force relation to single and repeated sprint swimming performance
Comment 1
2-3 – Please format the title text considering the journal template and journal instructions for authors (normally upper case requested).
Response
hank you for the comment. This was corrected as you suggested.
Comment 2
6-9 – Please include zip codes.
Response
This was corrected as you suggested.
Comment 3
25- Keywords should be presented in lower case and separated by “;”.
Response
Thank you for the comment. This was corrected as you suggested.
Comment 4
34-35 – Please format the manuscript considering the journal template, with no space between paragraphs.
Response
Thank you for the comment. This was corrected as you suggested.
Comment 5
51-52 – Please consider abbreviating the kinematic variables, as in the abstract.
Response
Thank you for the comment. This was corrected as you suggested.
Comment 6
66 – Please describe all inclusion and exclusion criteria.
Response
Swimmers were free from injury, and no medication was used prior or during experimental procedures also were required to have at least 5 years of swimming in competitive level (lines: 75-77)
Comment 7
67 – Please characterize the swimmers, weekly training routines for example.
Response
All swimmers participated in 6 swimming training sessions, and one or two dry-land sessions per week. This information is now included in the “participants” paragraph (lines 76-77).
Comment 8
73 - Familiarization process by the subjects? Refrain from intense training previously to data collection?
Response
Thank you for the comment. We have inserted information concerning familiarization session (lines 96-100). Also, we include the information about swimmers previous training (line 75-77).
Comment 9
74 – Please describe in detail all the procedures. Some examples: swimming pool characteristics (size, number of lanes and lanes for data collection, deep, air and water temperature, humidity), dryland conditions, and criteria (space, number of attempts, and registration criteria). Moreover, who collected the data? Academic background, experience? All these and other details are very important to be clearly understood by readers.
Response
Thank you for the comment. Academic background of researchers is presented in lines 95-96. Details concerning swimming pool, indoors gymnasium and ambient conditions have been included in lines 92-95.
Comment 10
85 – Please revise the figure content. For example, session 3 is not correctly formatted.
This was corrected as you suggested.
Comment 11
96 – City of the instrument is missing.
Response
This was corrected as you suggested (line: 118)
Comment 12
96-97 – “F-V” and “P-V” were previously abbreviated.
Response
This was corrected as you suggested.
Comment 13
101 – Please confirm if the letter type and size of the figures correspond to the journal template.
Response
This was corrected as you suggested.
Comment 14
137 – Please indicate sample power and normality test.
Response
The information has been inserted in line 161.
Comment 15
Page 5 – Some topics and subtopics are in upper and others in lowercase, please standardize considering the journal template and journal instructions for authors.
Response
This was corrected as you suggested
Comment 16
156-158 – Please correct the tables and figures’ title and footnote format, considering the journal template and journal instructions for authors.
Response
This was corrected as you suggested
Comment 17
169, 178 – Please avoid starting the phrases with abbreviations.
Response
Thank you for the comment. This was corrected as you suggested.
Comment 18
171 – Please include space “100,200”.
Response
This was corrected as you suggested.
Comment 19
190 – Please revise all tables’ format and content.
Response
Thank you for the comment. This was corrected as you suggested.
Comment 20
218 – Please indicate the p values (in all manuscript, italic is suggested).
Response
This was corrected as you suggested.
Comment 21
236 – “Morouço”, please correct.
Response
This was corrected as you suggested.
Comment 22
255 – Kinematic variable previously abbreviated. Please revise these details throughout the manuscript.
Response
Thank you for the comment. This was corrected as you suggested.
Comment 23
265 – Please develop the limitations section and include suggestions for future research.
Response
Thank you for the comment. Limitations section and suggestions for future research was included in lines 309-315.
Comment 24
298 – Please correct the references format considering the journal template and journal instructions for authors.
Response
Thank you for the comment. This was corrected as you suggested.
Reviewer 2 Report
Dear Authors,
I would like to express my gratitude for the opportunity to review this manuscript.
At this stage, the document requires improvements, below with line indication:
2-3 – Please format the title text considering the journal template and journal instructions for authors (normally upper case requested).
6-9 – Please include zip codes.
25- Keywords should be presented in lower case and separated by “;”.
34-35 – Please format the manuscript considering the journal template, with no space between paragraphs.
51-52 – Please consider abbreviating the kinematic variables, as in the abstract.
66 – Please describe all inclusion and exclusion criteria.
67 – Please characterize the swimmers, weekly training routines for example.
73 - Familiarization process by the subjects? Refrain from intense training previously to data collection?
74 – Please describe in detail all the procedures. Some examples: swimming pool characteristics (size, number of lanes and lanes for data collection, deep, air and water temperature, humidity), dryland conditions, and criteria (space, number of attempts, and registration criteria). Moreover, who collected the data? Academic background, experience? All these and other details are very important to be clearly understood by readers.
85 – Please revise the figure content. For example, session 3 is not correctly formatted.
96 – City of the instrument is missing.
96-97 – “F-V” and “P-V” were previously abbreviated.
101 – Please confirm if the letter type and size of the figures correspond to the journal template.
137 – Please indicate sample power and normality test.
Page 5 – Some topics and subtopics are in upper and others in lowercase, please standardize considering the journal template and journal instructions for authors.
156-158 – Please correct the tables and figures’ title and footnote format, considering the journal template and journal instructions for authors.
169, 178 – Please avoid starting the phrases with abbreviations.
171 – Please include space “100,200”.
190 – Please revise all tables’ format and content.
218 – Please indicate the p values (in all manuscript, italic is suggested).
236 – “Morouço”, please correct.
255 – Kinematic variable previously abbreviated. Please revise these details throughout the manuscript.
265 – Please develop the limitations section and include suggestions for future research.
298 – Please correct the references format considering the journal template and journal instructions for authors.
Please carefully revise the English in the manuscript and the document format.
Minor editing of English language required.
Author Response
Response to Reviewer 2
Revised Manuscript ID: JFMK- 2511252. Paper title: Dry-land force–velocity, power-velocity and swimming specific force relation to single and repeated sprint swimming performance
The objective of this study was to determine the relationships of dry land and in water force variables with swimming performance and kinematics during sprint, middle distance and repeated swim sprints. The paper considers a very significant topic for sport science and coaching, yet the manuscript must be improved to be considered for print. First of all the English must be improved as the style is rather poor and some of the terminology is not correct and must be changed.
Response:
Thank you for the positive comments. We have tried to improve English and terminology across the text.
The introduction is a little short and the research objective should be better justified. Considering the very low sports level of the study participants the number of subjects included in the study is quite low. National level 16-17 year old swimmers reach times below 60s in the 100m and the studied population reaches times of close to 70s, which is recreational swimming. Considering the low sports level of the participants it may be questionable whether they has been introduced to systematic resistance training on land and in the water. This could affect the results of the study and unless the participants have a history of strength and power training behind them, the implications from the study are of little practical value.
Response:
We have inserted text to justify the research objectives (lines 36-42).
The level of the swimmers in our study is indicated in Table 1 by FINA point system. The participants may be characterized as national level swimmers according to the categorization as proposed by McKay, et al., Int. J. Sports Physiol. Perf. 2022, 17, 317–331.https://doi.org/10.1123/ijspp.2021‐0451). Despite the moderate competitive level of swimmers in front crawl, they have at least two years of experience in dry-land resistance training. We agree with the reviewer that our findings should not be generalized, and we have included a limitation section (lines 310-315).
The materials and methods section is written rather w)ell, with the study design presented clearly in figure 1. I do not understand the difference between 1 RM in the bench press and maximum strength (MS).
Response:
This has been clarified in line 111. 1 RM was used as an index of maximum strength (MS).
I also have some doubts about the possibility of conducting 4 all out tests at 50, 100, 200 and 400m in the same testing session, considering the very low sports level of the participants, thus fatigue could have impacted the longer time trials.
Response:
Αs mentioned in the text and in figure 1, the tests were applied by swimmers on two different days. On the first day the 50 and 100 m tests and on the second day the 200 and 400 m tests, all with a 30 min resting interval between tests.
The results are presented rather well in 3 tables, with some technical issues in table 4. The authors should clearly state that the negative correlations of tethered force or impulse with swimming performance indicate that a higher value of these variables go along with better performance in swimming which means lower time values.
Response:
We have clarified this issue in lines 175-176 of the revised manuscript.
The discussion is perhaps the best part of the manuscript, yet it should compare the obtained results to similar studies. The main finding is that both dry land bench press performance and in water force variables significantly correlate with swimming performance. The low sports level of the participants, as in elite swimmers non-specific dry land strength exercises do not correlate with swimming performance, while only water specific strength and power variables affect swimming performance, may influence these results.
Response:
We thank you for this comment. Unfortunately, we are not aware of studies using elite swimmers to test such relationship. Most of the studies have used moderate level and young or children swimmers (Garrido, N et al., Journal of Human Sport and Exercise 2010, 5 (2), 240–249; Pérez-Olea, et al., Journal of Strength and Conditioning Research 2018, 32 (6), 1637–1642). It is likely that in a homogenous group of elite swimmers other factors, such as swimming efficiency, may dictate performance.
There are only 23 references chosen yet they are not used properly in explaining the results, or in justifying the objective of the study. Consider adding some of the recent empirical studies related to the topic, which may help in confronting the obtained results with those of other authors. I suggest revising the manuscript and resubmitting it for additional evaluation. Please check the English carefully to use proper terms acceptable in sport sciences.
Response:
We have added relevant references to support the objectives of the study (# 18 and # 22). We have check English and terminology across the text according to your suggestion.
Round 2
Reviewer 2 Report
Dear Authors,
Thank you for considering my suggestions and incorporating them into the manuscript, which globally improved, congratulations.
Below suggestions related to this last version (v2), with line indication.
2-3 – Please format the title text considering the journal template and journal instructions for authors (normally upper case requested, only at the beginning of the word).
6-9 – Please format the text considering the journal template and instructions for authors (city, country, emails, and initials of the authors).
32 – Please remove the space between citation numbers.
59 – In the first appearance in the manuscript, the word should be in full, followed by the abbreviature. Afterward, only the abbreviature.
104 - Please revise the figure content. For example, session 3 is not correctly formatted.
123 - Please confirm if the letter type and size of the figures correspond to the journal template.
161 - Please indicate sample power (GPower calculations and outputs).
218 - Please format the tables, considering the journal template and journal instructions for authors.
261 – “4x50” the unit is missing.
328 – Please delete the line.
345 – Please correct the references format considering the journal template and journal instructions for authors. For example, journals should be abbreviated.
Please carefully revise the English in the manuscript and the document format.
Minor editing of English language required.
Author Response
Response to Reviewer 2
Revised Manuscript ID: JFMK- 2511252. Paper title: Dry-land force–velocity, power-velocity and swimming specific force relation to single and repeated sprint swimming performance
Dear Authors,
Thank you for considering my suggestions and incorporating them into the manuscript, which globally improved, congratulations.
Below suggestions related to this last version (v2), with line indication.
Response:
Thank you for the positive comment. We have tried to improve our manuscript according to your suggestions.
2-3 – Please format the title text considering the journal template and journal instructions for authors (normally upper case requested, only at the beginning of the word).
Response
Thank you for your comment. This was corrected as you suggested.
6-9 – Please format the text considering the journal template and instructions for authors (city, country, emails, and initials of the authors).
Response
This was corrected as you suggested (lines: 6-10).
32 – Please remove the space between citation numbers.
Response
This was corrected as you suggested (line 33).
59 – In the first appearance in the manuscript, the word should be in full, followed by the abbreviature. Afterward, only the abbreviature.
Response
This was corrected as you suggested (lines 58-59).
104 - Please revise the figure content. For example, session 3 is not correctly formatted.
Response
All Figures and Tables were changed according to the journal information for authors.
123 - Please confirm if the letter type and size of the figures correspond to the journal template.
Response
We have changed the letter type to palatino linotype 10 according to the journal template. Also, we confirm that the figure size corresponding to the journal template.
161 - Please indicate sample power (GPower calculations and outputs).
Response
Thank you for your comment. We have included this information in lines 159-161.
218 - Please format the tables, considering the journal template and journal instructions for authors.
Response
The tables changed according to journal instructions.
261 – “4x50” the unit is missing.
Response
This was corrected as you suggested (line 262).
328 – Please delete the line.
Response
This was corrected as you suggested.
345 – Please correct the references format considering the journal template and journal instructions for authors. For example, journals should be abbreviated.
Response
The references’ section was corrected according to journal instructions.
Please carefully revise the English in the manuscript and the document format
Response
We have made an effort to revise our English in the manuscript, especially in the discussion section.